# Fostering Broad Oral Language Skills in Preschoolers from Low SES Background

**DOI:** 10.3390/ijerph17124495

**Published:** 2020-06-23

**Authors:** Raffaele Dicataldo, Elena Florit, Maja Roch

**Affiliations:** 1Department of Development and Socialization Psychology, University of Padova, 35131 Padova, Italy; maja.roch@unipd.it; 2Department of Human Sciences, University of Verona, 37129 Verona, Italy; elena.florit@univr.it

**Keywords:** narrative-based intervention, preschool, oral language development, socioeconomic status, classroom-based intervention

## Abstract

Socioeconomic disparities increase the probability that children will enter school behind their more advantaged peers. Early intervention on language skills may enhance language and literacy outcomes, reduce the gap and, eventually, promote school readiness of low-SES (Socioeconomic Status) children. This study aimed to analyze the feasibility and effectiveness of a brief narrative-based intervention (treatment vs. control group) aimed to foster broad oral language skills in preschoolers (N = 69; Mean age = 5.5, SD = 4 months) coming from low-SES families. Moreover, it was analyzed whether children’s initial vocabulary mediates the intervention’s responsiveness. Results have shown that children in treatment group obtained greater gains than children in control group in almost all intervention-based measures. There is also some evidence for the generalizability of the intervention to other skills not directly trained during the intervention. Moreover, it was found that children’s initial vocabulary mediates the intervention’s responsiveness showing that children with high vocabulary made greater gains in higher-level components of language comprehension, whereas children with low vocabulary made higher gains in vocabulary. Taken together, our findings suggest that a relatively brief, but quite intensive narrative-based intervention, may produce improvements on broad oral language skills in preschoolers from low-SES backgrounds.

## 1. Introduction

The Emergent literacy approach conceptualizes the acquisition of literacy as a developmental continuum with its origins early in the life of a child [1]. The development of adequate literacy, broadly defined as the ability to read and write texts [2], is pivotal for later school readiness and academic success [1]. Emergent literacy skills established in preschool, namely phonological awareness, vocabulary and narrative skills lay an important foundation for later reading comprehension [3]. These broad language skills serve as a bridge between oral and written language [4]; thus, children who enter school with inadequate oral language skills face significant challenges learning to read and comprehending what they hear and read [5].

Between the ages of 2 and 6, children show both large variation and rapid growth in oral language skills [6]. Differences in oral language skills appear to be relatively stable, suggesting that children with the poorest skills remain disadvantaged throughout their lives [6,7]. Attention to broad oral language skills during preschool years is imperative for the prevention of reading difficulties and academic failure [8,9]. Without early high-quality systematic language intervention, early children’s language difficulties may persist in elementary school and lead to reading difficulties and academic failure [10].

These findings have implications for interventions for children coming from low-SES backgrounds who generally struggle with oral language skills. Children coming from low-SES families show lower levels of oral language skills if compared with peers from more advantaged backgrounds on measures of language processing, comprehension and production [11]. The most striking evidence of SES disparities is observed in children’s expressive and receptive vocabulary: Children coming from high-SES families have larger expressive vocabularies compared to peers coming from low-SES families [12,13]. Early gaps in measures of language skills for socioeconomically underprivileged children persevere, producing an effect on the “readiness to learn”, and amplify as children progress through school [14].

Researchers report promising findings from interventions in educative settings using narratives as a tool for supporting the development of broad oral language skills [15]. The use of narratives in the form of shared book reading or storytelling is quite common in early childhood educational settings: These methods provide teachers with opportunities to scaffold children’s learning and language development [16]. Given the importance of narrative skills, including storytelling and story comprehension, for reading and reading comprehension [17,18], interventions based on narratives are well suited for preschool classrooms [19]. 

Collins (2016) analyzed the effects of low- and high- cognitive demand discussion about narratives, aimed to support inferential thinking during shared book reading, on preschoolers’ story comprehension [20]. Results have shown that high-cognitive demand talk during discussion has a positive effect on children’s comprehension. Thus, prompting children to engage in inferential thinking seems to foster their understanding of high-demand queries and enables them to generalize this thinking to new questions about the book. These findings suggest that the use of narratives, in the form of discussion after storybook reading, is a promising context for fostering inferential thinking as well as other comprehension-related abilities [20]. 

To sum up, it is suggested that one of the best ways to promote improvements in oral language skills and to prevent the increase in initial gaps over time is to target a broad set of skills early in development [21,22]. In the current study, we developed an intervention based on narratives intended to enhance a large set of oral language skills of preschoolers coming from families with low-SES, residing in north-eastern Italy. We evaluated the effectiveness of and children’s response to our intervention, analyzing the improvements following the intervention in two levels of abilities, measured through experimental materials related and standardized tasks, to address both near and far transfer effects, keeping in account initial differences in vocabulary size. 

### 1.1. Interventions Targeting Oral Language Skills During Preschool

Past studies have identified the last year of preschool as a crucial developmental step for school readiness, in which children’s narrative competence is linked to future literacy skills [23,24]. In the following paragraphs, we report evidence from previous studies investigating the effectiveness of interventions based on narratives’ use, promoting oral language components of narrative skills in preschoolers and specifically targeting children coming from low SES families.

#### 1.1.1. Interventions Targeting Narrative Skills in Preschoolers

Narrative skills, according to the multicomponent model of comprehension [25], involve a large range of oral language skills, such as vocabulary knowledge, morphosyntactic skills, inferential abilities, integration of previous knowledge, use of story schemas and metacognitive abilities, that developing hand in hand and interacting with the others have the potential to give rise to individual differences [21,26]. While all these skills support written comprehension, they have received very limited attention in intervention research with beginning readers [27,28] or prereaders [29]. 

Several meta-analyses have summarized the effects of interventions targeting narrative skills and related skills in preschoolers [30,31,32]. Numerous examples of interventions aimed to foster preschoolers’ broad oral language skills have been suggested [31]. Three main approaches to foster oral skills have been used: (a) dialogic book reading, (b) direct teaching of language comprehension skills and (c) an approach that combines book reading, vocabulary instruction and exercises for language comprehension-related skills [33]. A meta-analysis examining the effects of dialogic book reading showed moderate effects on expressive and receptive vocabulary [32]. Another meta-analysis examining the effects of vocabulary instruction showed moderate to strong effects for experimental vocabulary measures and small to moderate effects on standardized vocabulary tests [30]. Overall, numerous studies have shown positive effects for vocabulary learning in monolingual preschoolers [34,35,36], as well as in bilingual children [37,38], confirming the benefits of these interventions for preschoolers. However, although narratives are the best tool for supporting the development of broad oral language skills, most narrative programs that have been developed to foster oral language skills have focused just on phonological awareness and vocabulary outcomes [39,40,41,42], whereas few extended their analysis on other oral language skills. 

In 2010, Spencer and Slocum developed an intervention curriculum named “Story Champs” aimed to promote children’s narrative competence, focused on story grammar and complex language features used when telling stories [43]. Results of numerous studies using this curriculum show significant gains in children’s narrative competence. Narrative instruction yielded statistically significant improvements with moderate to strong effect sizes (ranging from 0.49 to 1.26) for the children’s story comprehension and retelling skills when compared with children in the control group [44]. 

Within the Italian context, it was developed and tested an intervention targeting narrative production for preschool children focused on enhancing genre knowledge, macrostructure (structure and coherence) and microstructure (cohesion) [23]. Findings show that the effect of the intervention was higher on structure and coherence, whereas growth in cohesion was characterized by smaller effect size, suggesting a higher efficacy of narrative interventions on macro-structural levels, rather than micro-structural ones [23]. A recent meta-analysis on 15 studies on narrative instruction in preschool, showed that this kind of interventions improved expressive and receptive aspects of language, namely temporal sequencing, story grammar knowledge, literal recall and inference making ability [45]. Most of these studies were focused on narrative abilities in production, whereas very few studies have been interested in oral narrative comprehension. Overall, the literature reported that narratives are a promising context for fostering simultaneously a wide range of linguistic and cognitive skills in preschool children [20].

#### 1.1.2. Intervention Targeting Oral Language in Low SES Preschoolers

Despite the evidence for the effectiveness of different narrative-based interventions, questions remain open about what interventions’ features are likely to reduce the chances that preschoolers at risk of later reading problems will experience them. Children from low-SES backgrounds are most likely to be at risk for inadequate language growth [2,9,46,47]. There is, thus, clearly a need to examine the effects of narrative-based interventions in promoting broad oral language skills in preschoolers coming from low SES backgrounds that generally present low oral language skills [11,48]. The knowledge about how to best help low-SES preschoolers to develop broad oral language skills is still sparse and limited [19]. To date, very few narrative-based interventions for young children with language difficulties, such as children coming from low SES families, exist [49,50]. 

Torgesen (2000) has highlighted in his review the importance of research in the area of intervention and argued that the two abilities that should be targeted in preschool intervention programs to prevent later literacy difficulties associated with low SES backgrounds are decoding and oral language skills [51]. However, research supports limited understanding about how to improve broad oral language skills, before formal instruction, in children coming from low SES background. Nearly all researches on interventions with low SES children focused on oral vocabulary [52], whereas only a smaller number of works have examined narrative-based interventions for their contribution to other oral language skills, such as inferential abilities and narrative comprehension. Spencer and colleagues (2015) investigated the effect of Story Camps intervention on a large group of low SES preschoolers on several oral language skills using narrative production and comprehension tasks [19]. Results have shown improvements in story retell and story comprehension measures for the treatment group compared to the control one and no effects on personal story generation measures, namely the children’s ability to tell a brand-new story without a preexistent script to follow. 

The current study addressed the need to examine the effects of narrative-based interventions in promoting broad oral language skills in preschoolers coming from low SES backgrounds by examining the effects of a short narrative-based intervention on broad oral language skills namely vocabulary, inferential abilities, knowledge of story structure and narrative comprehension.

### 1.2. Response to Intervention

An additional issue also considered in this study concerns the individual differences in the responsiveness to intervention. Response to Intervention (RtI) is a framework for identifying children with emerging difficulties and leads to the subsequent provision of differentiated instruction according to individual children’s needs [19].

An increasing number of studies suggest that children respond differently to intervention based on their prior knowledge. Particularly, in language-focused interventions, children respond differently to intervention based on their initial vocabulary knowledge, an indicator of prior knowledge [53]. Usually, according to the phenomena called the Matthew effect, whereby advantages and disadvantages accumulate, so that “the rich get richer and the poor get poorer” [54], children with stronger language make the greatest gains after interventions [55,56]. However, explicit and sustained instruction may attenuate or eliminate this effect allowing, even children with lower initial levels of vocabulary, to make greater gains after a language-focused intervention [57].

### 1.3. The Current Study

In this study, conducted with an emergent literacy approach [1], assuming a long-term perspective oriented to facilitate school readiness and prevent later difficulties with reading comprehension for preschoolers coming from low SES families, we developed and analyzed feasibility and effectiveness of a brief narrative-based group intervention (8 weeks) aimed to foster broad oral language skills. The intervention that we validate in this study extends previous literature in narrative competence interventions as focuses on enhancing, through different activities, the use of inferential abilities to infer the meaning of words, story structure and information from texts. The main contribution, as compared to previous studies, is twofold: First, we focus on comprehension skills, whereas most of the studies targeting narrative skills focused on expressive and production skills [15,44]. Second, most of the existing studies focused on the components of narrative competence individually [39,40,42], whereas in the current work we addressed different components of broad language comprehension altogether.

In this study, we targeted children from low SES backgrounds because a vocabulary weakness may constrain the ability to generate inferences, which in turn, limit the ability to infer the meaning of novel words [58] and hinder oral comprehension of narratives. Furthermore, weak oral language skills may produce an effect on the “readiness to learn” and amplify the children gap in a negative cycle, once they enter school.

### 1.4. Research Questions

This study aims to answer the following research questions: 

(1) Is this brief narrative-based group intervention combining shared book reading, vocabulary instruction and exercises on different narrative skill components, effective in fostering broad oral language skills in preschoolers coming from low-SES families? Do children take part in the intervention improved their oral language skills more than children in the control group?

We analyzed the improvements following the intervention in two levels of abilities, measured through related experimental materials (i.e., intervention-based measures) and standardized tasks to address both near and far transfer effects. Higher improvements in all abilities were expected for children taking part in the intervention than for children who did not participate in the intervention activities. 

(2) Do children respond differently to the intervention based on their initial level of vocabulary? Does the vocabulary knowledge affect children’s improvement? 

It was expected that children with higher vocabulary, because of the Matthew effect, would obtain greater gains after the intervention than peers. In particular, it was expected a higher improvement in inferential abilities due to the relationship between vocabulary and inferential abilities [58]. However, our main purpose was to meet the needs of less advanced children by providing sufficient support to enable those with lower initial levels to benefit from the intervention and, according to a compensatory model, to reduce the initial differences over time, counter the Matthew effect.

## 2. Materials and Methods

### 2.1. Participants

This project, approved by the Ethics Committee of the Representative Institution (1639/Univeristà Degli Studi di Padova), was carried out in a preschool in the metropolitan area of a medium-sized city in the Northeastern of Italy. In the Italian school system, preschool includes 3 to 5-year-old children. Parents were informed about the project during a parent-researchers meeting, signed a written consent form to let their child participate. A group of sixty-seven children (37 female) coming from low-socioeconomic families, aged between 4 years and 11 months to 6 years and 1 month (Mean age = 5.5, SD = 4 months) participated to the study. 

Low socioeconomic status was determined through a combination of the measure of the educational level of both parents and annual family income. Information collected through a questionnaire administered to parents (mean age 38) showed that 78% of mothers and 83% of fathers had high school diplomas or less (from 5 to 13 years of education), whereas only 22% of mothers and 17% of fathers had pursued post-secondary studies. These latter percentages are lower than the national average (30% of Italian females and 22% of males between the ages of 35 and 39 pursue post-secondary studies). Concerning annual family income, from eighty-seven percent of parents (N = 61) who agreed to give this information, the majority (N = 52) declared an annual family income below 34,000 € that represents the National mean income [59]. In detail, 15 parents declared an annual income below 24,000 €, 16 between 24,000 € and 30,000 € and 21 between 30,000 € and 34,000 €. Overall, the socio-economic information collected through the questionnaire (i.e., education level of both parents and annual family income), converge in showing that children involved in this study come from medium-low to low socioeconomic-status background.

### 2.2. Research Design and Procedure

A quasi-experimental research design with pre-test/post-test comparison group was employed for this study. Two classes were assigned to treatment (N = 50) and one to the comparison condition (N = 17). In both groups, participants’ receptive vocabulary, inferential abilities and narrative comprehension were assessed before implementing intervention with the treatment group (pre-test) and immediately following the completion of the 8-week treatment phase (post-test), with intervention-based measures and standardized tests to address both near and far transfer effects. Six trained master students individually administered all tasks in a fixed order: None of them was aware of the children’s group assignment. Each child was tested over two sessions of 30 min.

### 2.3. Materials

#### 2.3.1. Intervention-Based Measures (Near Transfer Effects)

We developed three experimental probes to verify whether children benefited directly from each intervention activity described above, thus to assess near transfer effects of the intervention.

1. Probes targeting Activity 1: Inferring words meaning.

To verify if the children learned the challenging five target words used during the intervention (mist; joker; sparkle; incautious and particolored), we developed a sentence completion task and a word recognition task. 

During the sentence completion task, the examiner read brief sentences describing a situation in which the target word could be inserted and the children were asked to complete the sentence. The context presented in the sentences was different from the original context and from the other contexts used during the intervention; thus, the children must have generalized the meaning of the new word learned to complete the sentences. Answer to each sentence was evaluated on a 0–1-point scale: The incorrect answer was scored 0, whereas correct was scored 1 (range 0–5). The reliability, evaluated by calculating Cronbach’s alpha over the five items’ scores at Time 2 was 0.67

During the word recognition task, children were asked to indicate which out of three pictures best represent the word pronounced by the examiner (target picture, semantic distractor and phonologic distractor). The five target words and the ten-filler words were randomly presented to children. Each item concerning target words was evaluated on a 0–1-point scale: The incorrect answer was scored 0, whereas correct answer was scored 1 (range 0–5). The reliability, evaluated by calculating Cronbach’s alpha over the five items’ scores at Time 2 was 0.56. This task was delivered before and after the intervention to the treatment group, while it was presented to the control group at the two-time points.

2. Probes targeting Activity 2: Inferring temporal and causal links.

To verify the children’s textual (information contained within the story) and inferential comprehension (information about the story that has to be inferred from the story) of the story used during Activity 2, we developed a task in which participants were asked to point if the statement about the story was true or false. The task was focused on the two types of information necessary to understand the story: textual (18) and inferential (18). The same number of true and false statements were provided, namely 18 of each type. Answer to each item was evaluated on a 0–1-point scale: The incorrect answer was scored 0, whereas correct answer was scored 1 (range = 0–36). Two separate scores (range = 0–18), one for textual and one for inferential answers were calculated. The reliability, evaluated by calculating Cronbach’s alpha over the two scores, was 0.81 at Time 1 and 0.83 at Time 2. This task was administrated before and after the shared dialogical book reading to the treatment group. The control group assisted in a passive reading of the book, and the task was administered to them both before and after the reading. 

3. Probes targeting Activity 3: Inferring the correct sequence of the story

To verify whether participants improved their ability to infer the correct order of pictured stories, we developed a task of story picture reordering. The task was to rearrange two sequences of stories in the correct order each composed by six pictures, to obtain a story of complete meaning. Two different scores were obtained: accuracy (0–12) and speed. The reliability, evaluated by calculating Cronbach’s alpha over the two accuracies’ scores at Time 2 was 0.59. The task was administered before and after the intervention to the treatment group, whereas it was administered to the control group at the two-time points without treatment.

#### 2.3.2. Standardized Tasks (Far Transfer Effects)

Receptive vocabulary (PPVT-R): The PPVT Revised is a standardized test that measures receptive vocabulary [60]. It consists of a list of words presented to participants who are asked to indicate which out of four pictures best represent the target word. The items are presented in order of increasing difficulty. A basal level is defined based on the child’s ability to give eight consecutive correct answers. Testing is then continued until the participant obtains six incorrect answers out of the last eight words presented (ceiling level). Raw scores correspond to the number of correct answers minus the number of errors. The reliability for the PPVT-R, which was evaluated using the split-half procedure, is 0.88.

Narrative text comprehension (TOR 3–8): The test TOR 3–8 is a standardized test for Italian children aged between 3 and 8 years of age that measures narrative text comprehension [61]. Participants are asked to listen to two stories appropriate for 4 to 6-year-old children. The tester read the stories aloud and, to minimize the cognitive and memory load, he/she interrupted reading at two predetermined points and asked multiple-choice comprehension questions. The tester presented four alternative answers both verbally and using pictures. Participants are asked to point to the correct picture. Comprehension was assessed for each story using 10 questions, half concerning information explicitly stated in the story and half requiring inferences to be generated. The score consists of the sum of correct answers, 10 for each story, with a maximum score of 20. The reliability for the TOR 3–8, evaluated by calculating Cronbach’s alpha over two items, was 0.76.

Inferential abilities task: Inferential abilities were assessed using an inference task used in a previous study [62]. The task consisted of ten items, each containing two short sentences read aloud referring to common and familiar events, followed by two inferential questions. The questions focused on two types of inferences: Knowledge-based and text-based inferences. The knowledge-based inferences require information from previously acquired world-knowledge to be incorporated within the episode (e.g., “That day Piero could not wait to put on the swimsuit to play with a scoop and a bucket. Where he had gone that day?”), instead, the text-based inferences are necessary to connect various pieces of information provided in the short episode and to identify their implicit relations (e.g., “Then Piero picked up the scoop and the bucket. He put the games in the bag; where are the scoop and the bucket?”). Answer to each question was evaluated on a 0–2-point scale: A completely incorrect answer was scored 0, whereas partially correct answer or answer provided after a clarification was scored 1 and a fully correct answer was scored 2. Three scores were calculated: Knowledge-based inferences (range = 0–20), text-based inferences (range = 0–20) and total inferences (range = 0–40). The reliability, evaluated by calculating Cronbach’s alpha over the items, was 0.54.

### 2.4. Treatment Fidelity and Control Condition

To ensure reliability in the appropriate delivery of the intervention, it was carried out by a Ph.D. student and two master students trained to deliver it, who followed written instructions previously prepared for each activity. Since the activities were carried out during school hours, to avoid behavioral changes in teachers, small groups of five or six children randomly created, were pulled out of class and accompanied to the room used for the activities. Each of the small groups formed on the first day of the intervention became a “team” that worked together for all sessions of intervention. Teachers of the comparison classroom (control group) continued to read to their class during this time in the schedule and were unaware of the narrative-based intervention procedures being taught to the treatment group.

### 2.5. Description of the Intervention Targeting Broad Oral Language Skills

The intervention, targeting broad oral language skills through narratives, combines shared book reading, vocabulary instruction, strategy instruction, questioning and discussion to support inferential thinking and exercises on story structures aimed to improve broad oral language skills in preschoolers. 

The narrative-based intervention consisted of eight weekly sessions, each lasting 45 min for a total of 6 hours of intervention. During each session, three different activities were delivered:

1. Inferring words meaning: Within the book “Che tempo fa?” [63] were inserted into five challenging and infrequent words. Each word—one per session—was repeated four times: (a) incidental exposure: children listen to part of the story in which the target word was inserted; (b) expansion and definition of the original context: children were questioned about the meaning and received the definition; (c) link to previous knowledge: the new word was inserted in a new context, familiar to children; (d) long-term recall: in the subsequent session children were asked to recall the word learned in the previous session.

2. Inferring temporal and causal links during shared book reading. The illustrated book “Il Litigio“ [64] was divided into different sections and each section was read aloud during one session. The reading, performed by two trainees sited in circle whit children, was interrupted in predetermined points to discuss temporal or causal inferences. For temporal inferences, the order in which two events were narrated in the story was discussed, whereas for causal inferences children were stimulated through questions to think and discuss on the causal connection between two events (e.g., “why do the two rabbits manage to escape?”, “because they’re digging together”) or between an event and a reaction of the story character. (e.g., “why does the fox feel defeated?”, “because she couldn’t catch the two rabbits”). The difficulty of inferences generation grows during each session: events to be connected become more and more distant in the story or become gradually more implicit to infer.

3. Inferring the story structure. For this activity, we used a new set of five pictured stories in each session. Children were asked to look carefully at the picture of the story and to re-arrange the sequences to obtain a story and tell the story obtained. If the children were unable to order the story correctly, after two attempts, the trainees helped them suggesting the starting image of the sequence.

## 3. Results

### 3.1. Preliminary Analyses and Descriptive Statistics

Table 1 shows descriptive statistics (means, standard deviations, minimum and maximum and variance) for treatment (TG) and control group (CG) and group comparison at Time 1. Standard scores are reported where available. 

Concerning performance on intervention-based measures, before the intervention, none of the children, both in TG and CG, was able to produce the target words in the sentence completion task and only a few children were able to recognize the target words, suggesting that choice of words was accurate. In the story comprehension task, after the first plenary reading, children performance covered a large range of scores with an average score of 25 out of 36 (no ceiling effects), whereas in picture story re-ordering task, children showed weak performance, rearranging on average three images out of 12. 

Concerning performance on standardized tasks, the average performance on receptive vocabulary (PPVT-R) lay at the lower boundary of the range appropriate for age. Children’s performance in narrative comprehension was age-appropriate and their performance in inferential abilities task covered a large range of scores.

Several independent-sample t-tests were run to determine if there were differences in performance at standardized, intervention-based tasks and chronological age between treatment and control group. There were no outliers in the data, as assessed by the inspection of boxplots. Whereas for the majority of the variables homogeneity of variances was not violated, it was violated, as assessed by Levene’s Test for Equality of Variances (*p* = 0.024) for performance on recognition task; thus, for this task in Table 1, *p*-value for equal variances not assumed is reported. Results show that there were no statistically significant differences between group means in all measures; thus, it is possible to assert that the two groups were well matched at Time 1. 

### 3.2. Intervention Effectiveness on Intervention-Based Measures (Near Transfer Effects) and Standardized Tasks (Far Transfer Effects)

To answer to the first research question, namely to examine the effectiveness of the intervention, we conducted a series of mixed 2 × 2 Analysis of Variance (ANOVA) with one between-subjects factor group (treatment vs. control) and one repeated-measure factor Time (pre-test, post-test) on each measure.

As it can be observed in Table 2, analyses of variance on intervention-based measures show several significant interactions group × time.

In particular, as far as intervention-based measures are concerned, in all cases except for comprehension of textual information, a significant interaction group × time was yielded indicating that the treatment group had greater gains than control participants in almost all intervention-based measures. 

Besides, as far as transfer effects are concerned, we found a significant group × time interaction on receptive vocabulary (PPVT-R) and narrative comprehension (TOR) measured through standardized tests (Table 3) indicating that the TG had greater gains than control participants.

### 3.3. Benefit Index

To better understand the effect size of the intervention, we calculated intervention gains for each participant as the difference between the improvement for the treatment group and the improvement for the control group divided by the standard deviation of the improvement from T1 to T2 for the group as a whole [65]. This enabled us to adjust the gains made by the treatment group to the gains made by the control group. The results, reported in Figure 1, indicated large effect sizes (over 0.80) on all standardized tasks and all intervention-based tasks, with the exception of word recognition. These results indicate that the treatment group showed relevant near effects as well as transfer effects of the intervention.

#### Intervention Effectiveness: Who Benefitted More From Intervention Activities?

To answer to the second research question, i.e., to examine whether children responded differently to the intervention based on their initial vocabulary, we calculated the intervention gains for each child of the TG as the difference between pre-and post-test performance, divided by the standard deviation at pre-test for the group of participants as a whole (i.e., benefit index).

Secondly, the treatment group was divided into three subgroups based on their vocabulary knowledge, measured trough the PPVT-R, at the pre-test. Children with an initial receptive vocabulary <89, namely children in the lowest tertile of the PPVT-R distribution, were classified as low vocabulary children (N = 17) and inserted in the Low Vocabulary Group (LVG), whereas children with an initial vocabulary higher than 101 (N = 17), i.e., higher tertile of the PPVT-R distribution, were inserted in the High Vocabulary Group (HVG). We compared intervention benefits for the two groups. As can be seen in Figure 2, both groups of participants benefitted from the intervention activities; however, differences between HVG and LVG were found in intervention benefits. 

To compare directly if the two groups benefitted differently from the intervention, Cohen’s d values were transformed into r indexes and then compared. The size of the differences, expressed in Cohen’s *q*, were reported in Figure 2 and interpreted according to Cohen’s guidelines: A difference  <0.1: No effect; from 0.1 to 0.3: Small effect; from 0.3 to 0.5: Medium effect; >0.5: Large effect. Concerning far effects, small differences were found in receptive vocabulary (*q* = 0.19), in favor of LVG, and inferential abilities (*q* = 0.19) in favor of HVG; no difference in the amount benefit between low and high vocabulary groups was found in narrative comprehension (TOR 3–8). Concerning near effects, a large difference was observed in story re-ordering task (*q* = 0.95), a medium difference was found in the word recognition task (*q* = 0.48) and a small difference in sentence completion task (*q* = 0.23) in favor of HVG, whereas small differences were found in intervention-based measures of story comprehension, namely textual questions (*q* = 0.10) in favor of LVG. 

## 4. Discussion

### 4.1. Intervention Effectiveness

The primary purpose of this study was to test the feasibility and the effectiveness of a brief narrative-based group intervention on broad oral language skills in preschoolers coming from low-SES backgrounds. Effectiveness of this intervention was analyzed both in terms of near transfer effects, using intervention-based tasks, and far transfer effects, using standardized tasks. Statistically significant differences between the treatment and control groups were found for all the intervention-based measures at post-test. These results demonstrated near effects of the intervention showing that activities and materials developed were adequate for preschool children and that they benefitted from these activities. Moreover, we found piecemeal evidence of generalizability of the intervention, namely treatment group showed greater gains than control group in vocabulary and narrative comprehension evaluated with standardized tests, whereas we did not find this result on the measure of inferential abilities. 

To our knowledge, no prior study has evaluated the possible synergistic effects of vocabulary instruction, strategy instruction, questioning and discussion to support inferential thinking and exercises on story structures. As this was one of the first examinations of a narrative-based group intervention involving preschoolers from low SES, the results of this study provide early support for its effectiveness. Our findings suggest that a relatively brief, but quite intensive intervention (6 hours in 8 weeks) based on narratives, can lead to sustained improvements on broad oral language skills in preschool children with low SES. Our results are in line with the previous literature on interventions targeting oral language skills that show short-term improvement in trained tasks [41,66,67] and in narrative comprehension [19,20]. However, only a few studies reported transfer effects to comprehension standardized tasks and even when generalized effects are reported, moderate to small effect sizes, which range from d = 0.16 to d = 0.29, are detected [52,68], or no effects [28]. Care should be taken in comparing effect sizes when measures of comprehension differ, as significant effects for intervention-based tasks are easier to obtain than effects on general measures of comprehension [20]. In the current study, we analyzed specific effects due to the intervention by calculating the intervention gains for each participant from the treatment group adjusting for the gains made by the control group. The results indicated that when compared to the control group, the trained group showed relevant near as well as far transfer effects of the intervention on all tasks showing effect sizes higher than 0.80. This preliminary result is promising in showing the effectiveness of this type of training delivered to low SES children. 

Different features of the intervention may be associated with children’s outcomes. For example, questioning promotes active child participation [69], elicits more responses from children than other forms of adult utterances [70] and promotes narrative comprehension when it includes inferencing [52,71,72]. Repeated readings and small-group instruction in preschool [34,73], including storybook reading contexts [74], contribute to children’s learning in preschool. In small-group (e.g., two to five) contexts, children are engaged in more cognitively challenging talk driven by children’s interests than in large-group settings [75]. Moreover, for comprehension, the opportunities for feedback and engaging in back-and-forth exchanges are likely to benefit inferential understanding [76].

### 4.2. Responsiveness by Language Knowledge

Our second research question is specifically related to the responsiveness to the intervention of preschoolers coming from low SES backgrounds. To this aim, it was further investigated to what extent children with high vocabulary and low vocabulary responded differently to the intervention. The analyses of effect sizes revealed that both groups of children, with high and with low vocabulary, benefited from the intervention, showing, however, benefits in different measures. Children with low vocabulary showed higher gains in receptive vocabulary (measure with a standardized task, PPVT-R) and in textual questions about the story (intervention-based task). Children with high vocabulary showed greater gains in the ability to use the story context for learning the target words (completion and recognition intervention-based task), in recognizing the correct story order (intervention-based task), but also generalized these skills by improving significantly their inferential abilities.

The significant contribution of baseline vocabulary to outcomes is consistent with other research showing that children with higher initial vocabulary make greater vocabulary gains than those with lower [35,37,56]; however, explicit and sustained instruction during the intervention may mitigate this effect. From our results, it seems that children with low vocabulary have benefited from the intervention in terms of “compensation”: They made gains in the ability in which they were weak, reducing the initial differences namely vocabulary, whereas they did not benefit in inferential abilities, at least partly, because their poor vocabulary level did not allow them to engage in high-level processes. Children with high vocabulary, on the other hand, improved more in their cognitive and integrative skills supporting the phenomena known as “Matthew effect” according to which, higher resources allow higher advantages from the learning process [77]. 

As argued by Silva and Cain (2015), the relationship between vocabulary and inferential abilities is reciprocal [58]. Better knowledge of words contained in the text facilitates the connections between different parts of the text and between the ideas expressed in the text and previous knowledge [78]. At the same time, inferential abilities facilitate new vocabulary acquisition, because the texts are the main source for new word learning from context. This may well represent a valid interpretation of our results concerning individual differences in the intervention benefits. In other words, even if all children involved in this study came from low SES backgrounds, they show different levels in initial receptive vocabulary and thus may have benefited differently from the intervention, according to their vocabulary size. Overall, these results are in line with our main purpose namely meet the needs of both less advanced and more advanced children by providing sufficient support to enable those with lower initial levels to benefit from the intervention and, at the same time, providing sufficient challenging opportunities for more advanced children. We can speculate that early intervention with children from a low level of SES—before the age of 4—should focus on vocabulary, to foster closing the gap with peers in this key component of language development, and then should move to other more complex oral language components of narrative comprehension, namely inferential abilities. 

## 5. Limitations and Future Directions

While the current investigation provides evidence that this intervention, delivered in small groups, is a promising intervention, the study is not without limitations. First, the assessment tools used to measure the direct effects of the intervention were researcher-developed tools. Secondly, since the small number of schools involved, the study design was weakened by our inability to randomly assign classrooms to treatment and control conditions. Finally, while the results highlight that children in the treatment group shown improvement after the intervention in several oral language skills, we have not monitored the medium/long-term maintenance of these effects. However, it is important to highlight that in this type of studies, going to investigate the effects in the medium to long term is very difficult because the participants are children with typical development and it is difficult to uniquely and specifically identify the effects of an intervention and distinguish it from natural development and learning.

This early efficacy study should be followed by replications involving more schools, using a stronger experimental design and analyzing long-term effects through a follow-up to examine the effects on school abilities. Despite the difficulties in evaluating the effectiveness of the intervention over time, repeated and close follow-ups over time could detect possible long-term effects of this kind of intervention.

There is a need for further studies aimed to develop and validate interventions, using narratives or storytelling, aimed to foster broad oral language skills in preschoolers. This effort to promote better oral language skills in preschool children and particularly in children from low SES backgrounds before the formal school could reduce early gaps in measures of language and give them the same opportunity of peers from high SES backgrounds to be “ready to learn” once they move the school and attenuate long-term SES-related effects.

## 6. Conclusions

From scientific review emerges that narrative-based interventions for pre-schoolers have typically targeted phonological and phonemic awareness and/or vocabulary, whereas a larger set of broad oral language skills including inferential abilities have rarely been considered [20]. To date, there are few studies showing benefits of narrative-based interventions for oral language skills in addition to vocabulary) in preschoolers [15,43], whereas there are more for school-aged children [28,29]. The findings of the current study provide some evidence that narratives represent a promising context for fostering broad oral language skills, namely vocabulary, inferential abilities, sequencing and in turn narrative comprehension, even in preschool children. The current work extends the existing literature on preschoolers lending preliminary convergent evidence in support of the results obtained by previous longitudinal studies [78]. 

Moreover, and most importantly, in this study, we focused on a specific population namely children from low SES backgrounds. Findings of numerous studies suggested that the different academic paths followed by children from different SES backgrounds have their roots in oral language skill differences established even before children start school [79,80]. However, if some language development trajectories have negative consequences for children’s literacy development and later school achievement, and if the trajectories with negative consequences have causes that can be remedied, perhaps efforts should be directed toward that goal. Since, as previously said narratives, in the form of shared book reading or storytelling, represent promising contexts for fostering broad oral language skills, those activities should be used as essential strategies in interventions aimed to shape developmental trajectories of broad oral language skills, especially in educational settings.

There are several potential benefits of embedding narrative experiences in preschool classrooms. First, all children, not just those who have significant language impairments, can benefit from narrative-based intervention. Second, teaching in a large group is more efficient and cost-effective than teaching one child or a few children at a time. Finally, children who may require more intense language intervention can be identified by assessing their response to narrative-based intervention provided to the whole class [19]. Our results, although preliminary are promising in showing the effectiveness of a narrative-based intervention aimed to foster broad oral language skills of preschoolers coming from low SES backgrounds. 

Questions of effectiveness and responsiveness are important, but if an intervention is not effective for diverse populations then it is severely limited. The observed improvements in vocabulary, inferential abilities and story comprehension are quite meaningful. Early interventions on language skills critical to enhance language and literacy outcomes may reduce the gap and, eventually, to promote school readiness of low SES children. Previous studies with low SES children have focused just on few language abilities and particularly on vocabulary since it represents their weakest oral skill. The current findings suggest that, although interventions’ gains might differ according to children’s initial levels of vocabulary knowledge, even other oral language skills may be trained in preschoolers from low SES backgrounds. 

## Figures and Tables

**Figure 1 ijerph-17-04495-f001:**
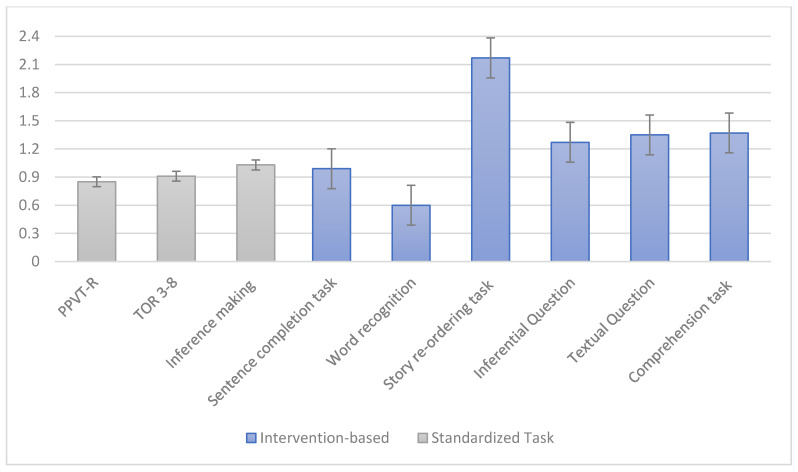
Standardized intervention gains for differences between pre-test and post-test.

**Figure 2 ijerph-17-04495-f002:**
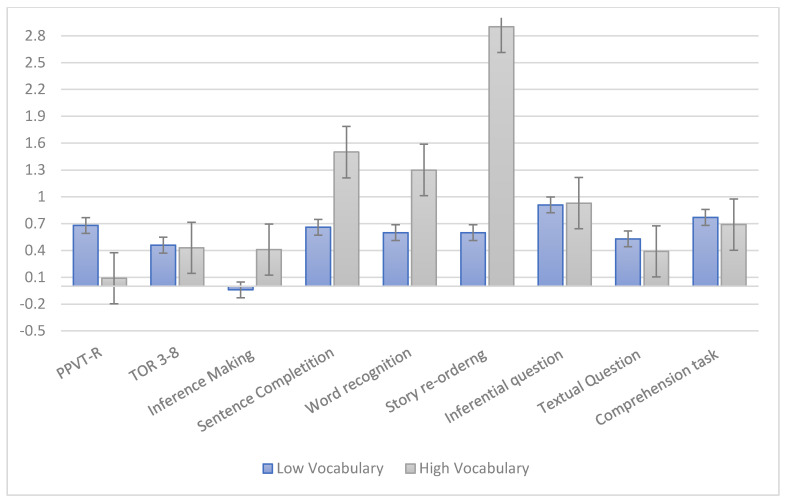
Standardized training gains for differences between pre-test and post-test for two groups (high and low vocabulary).

**Table 1 ijerph-17-04495-t001:** Characteristics of participants and group comparison at Time 1.

Variables	Treatment Group					Control Group					*t*-Test			
	N	Mean	SD	Range	Variance	N	Mean	SD	Range	Variance	*df*	*t*	*p*	*d*
Age	50	66.3	3.6	59–73	13.6	17	67	3.42	62–73	11.7	65	0.647	0.520	−0.20
Intervention-based measures														
Recognition task	40	2.5	1.4	0–5	2	17	3	1.17	0–5	1.3	55	1.30	0.199	−0.32
Comprehension task (total)	27	22.5	5.5	14–30	30.9	17	22.2	5	14–31	25.1	42	−0.171	0.865	0.01
Inferential questions	27	10.4	2.7	5–15	7.4	17	10.1	2.6	6–15	6.9	42	−0.277	0.783	0.12
Textual questions	27	12.11	3.3	7–17	11.2	17	12	2.7	8–16	7.5	42	−0.053	0.957	0.03
Story re-ordering task	13	3	2.1	0–7	4.7	17	2.5	1.6	0–7	2.6	28	−0.790	0.436	0.28
Standardized tasks														
PPVT (raw score)	50	79.5	22.2	21–136	495.5	17	72.7	24.2	18–107	587.4	65	−1.07	0.288	0.30
PPVT (standard score)	50	92.3	14.8	65–122	220.8	17	88	14.1	65–110	200.1	65	−1.03	0.303	0.30
TOR 3–8 (raw score)	50	13.1	3.2	3–17	15	17	11.8	2.6	8–16	7.1	65	−1.38	0.170	0.43
TOR 3–8 (standard score)	50	11.1	1.7	6–15	2.8	17	10.2	1.4	9–12	1.3	65	−1.82	0.073	0.56
Inferential abilities	50	19	6.6	6–35	44.3	17	17.5	6.5	5–28	42.8	65	−0.790	0.432	0.23

Note: PPVT = Peabody Picture Vocabulary Test; TOR 3-8 = Test di Comprensione del Testo Orale (Oral Text Comprehension Test); After the Bonferroni correction, the *p*-value was adjusted to 0.01.

**Table 2 ijerph-17-04495-t002:** Mean (SD) on proximal abilities at Time 1 and Time 2 and group comparisons.

	Treatment Group		Control Group		Anova Time × Group	*pƞ^2^*
	T1	T2	T1	T2		
Sentence completion task	0	0.85 (1.1)	0	0	T × G: F(1, 28) = 9.411 *	0.252
Word recognition	2.5 (1.4)	4 (0.98)	3 (1.1)	2.6 (1)	T × G: F(1, 55) = 17.321 **	0.240
Inferential questions	10.4 (2.7)	12.6 (2.8)	10.1 (2.6)	10.8 (2.3)	T × G: F(1, 42) = 5.515 *	0.116
Textual questions	12.1 (3.3)	14 (2.8)	12 (2.7)	13.3 (4.3)	T × G: F(1, 42) = 0.662	0.016
Comprehension task (total)	22.5 (5.5)	26.6 (5.3)	22.2 (5)	24.1 (6.3)	T × G: F(1, 42) = 4.353 *	0.094
Story re-ordering task (Accuracy)	3 (2.1)	8.3 (4)	2.5 (1.6)	4.8 (2.7)	T × G: F(1, 28) = 6.498 *	0.188

Note: * After the Bonferroni correction, the *p* value was adjusted to 0.008; ** *p* < 0.001.

**Table 3 ijerph-17-04495-t003:** Mean (SD) on distal abilities at T1 and T2 and group comparisons.

	Treatment Group		Control Group		Anova Time × Group	*pƞ^2^*
	T1	T2	T1	T2		
PPVT-R (Raw score)	79.5 (22.2)	88.2 (20)	72.7 (24.2)	74.8 (23.9)	T × G: F(1, 65)= 3.264	0.048
PPVT-R (std. score)	92.3 (14.8)	98.2 (14.9)	88 (14.1)	88.3 (15.3)	T × G: F(1, 65) = 5.323 *	0.076
TOR 3–8 (Raw score)	13.1 (3.2)	15.3 (2.2)	11.8 (2.6)	12.6 (3.3)	T × G: F(1, 65) = 3.968 *	0.058
TOR 3–8 (std. score)	11 (1.7)	11.5 (1.5)	10.2 (1.1)	10.5 (1.5)	T × G: F(1, 65) = 0.138	0.003
Inferential Abilities	19.1 (6.6)	22.6 (7.6)	17.5 (6.5)	19.1 (7.5)	T × G: F(1, 65) = 3.401	0.050

Note: * After the Bonferroni correction, the *p*-value was adjusted to 0.01.

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
