# Peer review of "Fostering Broad Oral Language Skills in Preschoolers from Low SES Background"

_ijerph, 2020, doi:10.3390/ijerph17124495_

Round 1

Reviewer 1 Report

This paper highlights an intervention study aimed at pre-schoolers from low SES backgrounds in northern Italy. The study adds to the existing body of research by extending and combining the focus to encompass comprehension skills, vocabulary, knowledge of story structure, and inference. The comparatively short study therefore included a comprehensive number of tests, which are well described and articulated. I appreciate that the authors seek to publish their success, and I also appreciate the self-highlighted limitation regarding lack of evidence of medium- and long-term impact of the study. As an ex-practising teacher, there is a cynic in me who will argue that these kinds of intervention are all well and good in theory, but in practice, they require considerations beyond the pure tests, linked to where they fit in the curriculum, whose role it is to run them, what gets squeezed out to make space for them, etc. With this comment, I do not wish to detract from the study itself (and, from my own experience when I feel a reviewer misses the point of the paper, I completely sympathise with the authors if they wish to make that point about me).

I am happy to recommend publication subject to minor revisions regarding the English language, or, depending on the editor's views, more major revisions, specifically linked to mid- and longer-term impact.

Author Response

Reviewer 1:

This paper highlights an intervention study aimed at pre-schoolers from low SES backgrounds in northern Italy. The study adds to the existing body of research by extending and combining the focus to encompass comprehension skills, vocabulary, knowledge of story structure, and inference. The comparatively short study therefore included a comprehensive number of tests, which are well described and articulated. I appreciate that the authors seek to publish their success, and I also appreciate the self-highlighted limitation regarding lack of evidence of medium- and long-term impact of the study. As an ex-practising teacher, there is a cynic in me who will argue that these kinds of intervention are all well and good in theory, but in practice, they require considerations beyond the pure tests, linked to where they fit in the curriculum, whose role it is to run them, what gets squeezed out to make space for them, etc. With this comment, I do not wish to detract from the study itself (and, from my own experience when I feel a reviewer misses the point of the paper, I completely sympathise with the authors if they wish to make that point about me).

I am happy to recommend publication subject to minor revisions regarding the English language, or, depending on the editor's views, more major revisions, specifically linked to mid- and longer-term impact.

RESPONSE: Thanks a lot for your comment, we are grateful to have received a comment from an ex-practicing teacher who, because of experience, appreciated the value of this study. As we highlighted in the limitation section, we have not monitored the medium/long-term maintenance of these effects that could have made this study more robust and attractive. However, it is important to highlight that in this type of studies, going to investigate the effects in the medium to long term is very difficult because the participants are children with typical development and it is difficult (or impossible) to uniquely and specifically identify the effects of an intervention and distinguish it from natural development and learning. In short, we fully agree with the point from a conceptual point of view, but we find it difficult to think and make it operational.

We agree with your cynical side about the difficulties in integrating this kind of intervention into the already busy preschools’ curriculum. However, we would like to highlight that this intervention has been carried out, as presented in this paper so that we can evaluate its effectiveness. Our aim was to see if the strategies used were effective in enhancing the language skills of pre-school children. Our desire was to suggest strategies that were effective and easy to apply in the classroom context during the shared reading activities typical of the preschool curriculum, at least in the Italian context, encouraging a shift from typical shared reading to dialogic reading, with frequent questions and greater active involvement of children during these activities. Therefore, the aim will be to provide teachers with instructions on how to apply these strategies in the classroom, having demonstrated their effectiveness of this small intervention through an experimental method and design.

We improved the Section “Limitations and future directions” with your suggestion (see line 548-593)

Reviewer 2 Report

I enjoyed reading the paper. The authors have presented an interesting article about Fostering broad oral language skills in preschoolers from low SES background. Despite the good work done, there is still some room for improvement, as follows:

- How do you think your findings can be useful applicable to other engineering disciplines?

Author Response

Reviewer 2:

I enjoyed reading the paper. The authors have presented an interesting article about Fostering broad oral language skills in preschoolers from low SES background. Despite the good work done, there is still some room for improvement, as follows:

- How do you think your findings can be useful applicable to other engineering disciplines?

RESPONSE: Dear Reviewer 2, thanks a lot for your comment. We are very pleased to read your appreciation for our work. It is important for us to receive this kind of feedback given the practical and applicative relevance of this work and the enormous difficulties in conducting this kind of studies.

As regards your very interesting question, this gave us time to reflect on possible future studies that we had already considered in the past. Our idea, in line with your question, was to digitize the material built for the intervention and then develop an App for mobile devices.

We believe that an interactive App that allows the child to receive visual and auditory feedback can be useful in different educational contexts. For example, in families with low SES or multilingual families where language knowledge may be limited, an App of this type, can provide appropriate language exposure and material for children. A child could use it independently or with the support of the parent who could provide more feedback and monitor the linguistic evolution of the child.

It could also be useful in school educational contexts or in clinical settings for the enhancement of language skills of children with limited language skills.

As said before, we tried to find funds for the development of an App starting from the results of this study, however, not having found them we decided to leave this idea in a drawer. Thank you for giving us the opportunity to rethink this idea for future projects. We hope to be able to realize it.

Reviewer 3 Report

This is an interesting idea and could be worked into a valuable series of studies. Although the research purpose and the statement of the problem are clear, it is difficult to exactly spot the research questions. The literature review is relevant. In order to improve the quality of the research, the following suggestions and comments to revise and refine the paper were stated:
1. Overall, the manuscript is well organized and easy to follow. The tables and figures are valuable in helping the reader interpret the findings.
2. As this is the first study of its type that I've seen in the discipline, it's certainly an original contribution.
3. Some references seem outdated, it is recommended to add some new ones. Some English sentences need to be adjusted to be more readable.

Author Response

Reviewer 3:

This is an interesting idea and could be worked into a valuable series of studies. Although the research purpose and the statement of the problem are clear, it is difficult to exactly spot the research questions. The literature review is relevant. In order to improve the quality of the research, the following suggestions and comments to revise and refine the paper were stated:
1. Overall, the manuscript is well organized and easy to follow. The tables and figures are valuable in helping the reader interpret the findings. 

RESPONSE: many thanks for this comment, we are pleased with your appreciation of the presentation of the results.

  1. As this is the first study of its type that I've seen in the discipline, it's certainly an original contribution.

RESPONSE: thank you for your comment and for appreciating the originality of the presented study. in order to exactly spot, within the paper, the research questions we have added the section Research questions (section 1.4). Here, we clarify our research questions and predictions.

  1. Some references seem outdated, it is recommended to add some new ones. Some English sentences need to be adjusted to be more readable.

RESPONSE: Dear Reviewer, we agree with you about the distant dating of some references and where possible we have tried to find more recent references. However, we would like to highlight that we wanted to mention some pioneering studies (see Whitehurst, G. J.; Lonigan, C. J. Child development and emergent literacy. Child Dev. 1998, 69(3), 848-872; Fazio, B.; Naremore, R.; Connell, P. Tracking children from poverty at risk for specific language impairments: A 3-year longitudinal study. J. Speech Lang Hear R. 1996, 39, 611-624; Hart, B.; Risley, T. R. Meaningful differences in the everyday experience of young American children. Paul H Brookes Publishing 1995; Stanovich, K. E. Matthew effects in reading: Some consequences of individual differences in the acquisition of literacy. Read. Res. Q. 1986, 21(4), 360–407.) for their relevance in the specific field.

Following your suggestion, we changed with new references:

[4] Marschark, M.; Spencer, P. E. The Oxford handbook of deaf studies, language, and education (Vol. 2). Oxford University Press. 2010

[9] Jackson, N. E.; Coltheart, M. Routes to reading success and failure: Toward an integrated cognitive psychology of atypical reading. Psychology Press. 2013. 

And removed [73]